# Learning to Perform Physics Experiments via Deep Reinforcement Learning

**Misha Denil**[1]    **Pulkit Agrawal**[2]    **Tejas D Kulkarni**[1]    **Tom Erez**[1]    **Peter Battaglia**[1]    **Nando de Freitas**[1,3]

[1]DeepMind    [2] University of California Berkeley    [3]Canadian Institute for Advanced Research

{mdenil,tkulkarni,etom,peterbattaglia,nandodefreitas}@google.com
pulkitag@berkeley.edu

## Abstract

When encountering novel objects, humans are able to infer a wide range of physical properties such as mass, friction and deformability by interacting with them in a goal driven way. This process of active interaction is in the same spirit as a scientist performing experiments to discover hidden facts. Recent advances in artificial intelligence have yielded machines that can achieve superhuman performance in Go, Atari, natural language processing, and complex control problems; however, it is not clear that these systems can rival the scientific intuition of even a young child. In this work we introduce a basic set of tasks that require agents to estimate properties such as mass and cohesion of objects in an interactive simulated environment where they can manipulate the objects and observe the consequences. We found that deep reinforcement learning methods can learn to perform the experiments necessary to discover such hidden properties. By systematically manipulating the problem difficulty and the cost incurred by the agent for performing experiments, we found that agents learn different strategies that balance the cost of gathering information against the cost of making mistakes in different situations. We also compare our learned experimentation policies to randomized baselines and show that the learned policies lead to better predictions.

## 1 Introduction

Our work is inspired by empirical findings and theories in psychology indicating that infant learning and thinking is similar to that of adult scientists (Gopnik, 2012). One important view in developmental science is that babies are endowed with a small number of separable systems of core knowledge for reasoning about objects, actions, number, space, and possibly social interactions (Spelke & Kinzler, 2007). The object core system covering aspects such as cohesion, continuity, and contact, enables babies and other animals to solve object related tasks such as reasoning about oclusion and predicting how objects behave.

Core knowledge research has motivated the development of methods that endow agents with physics priors and perception modules so as to infer intrinsic physical properties rapidly from data (Battaglia et al., 2013; Wu et al., 2015; 2016; Stewart & Ermon, 2016). For instance, using physics engines and mental simulation, it becomes possible to infer quantities such as mass from visual input (Hamrick et al., 2016; Wu et al., 2015).

In early stages of life, infants spend a lot of time interacting with objects in a seemingly random manner (Smith & Gasser, 2005). They interact with objects in multiple ways, including throwing, pushing, pulling, breaking, and biting. It is quite possible that this process of actively engaging with objects and watching the consequences of their actions helps infants understand different physical properties of the object which cannot be observed directly using their sensory systems. It seems infants run a series of "physical" experiments to enhance their knowledge about the world (Gopnik, 2012). The act of performing an experiment is useful both for quickly adapting an agent's policy to a new environment and for understanding object properties in a holistic manner. Despite impressive advances in artificial intelligence that have led to superhuman performance in Go, Atari and natural language processing, it is still unclear if these systems behind these advances can rival the scientific intuition of even a small child.

While we draw inspiration from child development, it must be emphasized that our purpose is not to provide an account of learning and thinking in humans, but rather to explore how similar types of understanding might be learned by artificial agents in a grounded way. To this end we show that we can build agents that can learn to experiment so as to learn representations that are informative about physical properties of objects, using deep reinforcement learning. The act of conducting an experiment involves the agent having a belief about the world, which it then updates by observing the consequences of actions it performs.

We investigate the ability of agents to learn to perform experiments to infer object properties through two environments—Which is Heavier and Towers. In the Which is Heavier environment, the agent is able to apply forces to blocks and it must infer which of the blocks is the heaviest. In the Towers environment the agent's task is to infer how many rigid bodies a tower is composed of by knocking it down. Unlike Wu et al. (2015), we assume that the agent has no prior knowledge about physical properties of objects, or the laws of physics, and hence must interact with the objects in order to learn to answer questions about these properties.

Our results indicate that in the case Which is Heavier environment our agents learn experimentation strategies that are similar to those we would expect from an algorithm designed with knowledge of the underlying structure of the environment. In the Towers environment we show that our agents learn a closed loop policy that can adapt to a varying time scale. In both environments we show that when using the learned interaction policies agents are more accurate and often take less time to produce correct answers than when following randomized interaction policies.

## 2 WHAT IS THIS PAPER ABOUT?

This is an unusual paper in that it does not present a new model or propose a new algorithm. There is a reinforcement learning task at the core of each of our experiments, but the algorithm and models we use to solve it are not new, and many other existing approaches should be expected to perform equally well if they were to be substituted in the same setting.

This paper is a step towards agents that understand objects and intuitive reasoning in physical worlds. Our best AI agents currently fail on simple control tasks and simple games, such as Montezuma's Revenge, because when they look at a screen that has a ladder, a key and a skull they don't immediately know that keys open doors, that skulls are probably hazardous and best avoided, that ladders allow us to defy gravity, etc. The understanding of physics, relations and objects enables children to solve seemingly simple problems that our best existing AI agents do not come close to begin to solve.

Endowing our agents with knowledge of objects would help enormously with planning, reasoning and exploration, and yet, doing so is far from trivial. What is an object? It turns out this question does not have a straightforward answer, and this paper is based around the idea that staring at a thing is not enough to understand what it is.

Children understand their world by engaging with it. Poking something to find that it is soft, tasting it to discover it is delicious, or hitting it to see if it falls down. Much of the knowledge people have of the world is the result of interaction. Vision or open loop perception alone is not enough.

This paper introduces tasks where we can evaluate the ability of agents to learn about these "hidden" properties of objects. This requires environments where the tasks depend on these properties (otherwise the agents have no incentive to learn about them) and also that we have a way to probe for this understanding in agents that complete the tasks.

Previous approaches to this problem have relied on either explicit knowledge of the underlying structure of the environment (e.g. hard-wired physical laws) or on exploiting correlations between material appearance and physical properties (see Section 7 for much more detail). One of the contributions of this paper is to show that our agents can still learn about properties of objects, even when the connection between material appearance and physical properties is broken. This setting allows us to show that our agents are not merely learning that blocks are heavy; they are learning how to check if blocks are heavy.

None of the previous approaches give a complete account of how agents could come to understand the physical properties of the world around them. Specifying a model manually is difficult to scale,

generalize and to ground in perception. Making predictions from only visual properties will fail to distinguish between objects that look similar, and it will certainly be unable to distinguish between a sack full of rocks and a sack full of tennis balls.

## 3  ANSWERING QUESTIONS THROUGH INTERACTION

We pose the problem of experimentation as that of answering questions about non-visual properties of objects present in the environment. We design environments that ask questions about these properties by providing rewards when the agent is able to infer them correctly, and we train agents to answer these questions using reinforcement learning.

We design environments that follow a three phase structure:

**Interaction**  Initially there is an exploration phase, where the agent is free to interact with the environment and gather information.

**Labeling**  The interaction phase ends when the agent produces a *labeling* action through which it communicates its answer to the implicit question posed by the environment.

**Reward**  When the agent produces as labeling action, the environment responds with a reward, positive for a correct answer and negative for incorrect, and the episode terminates. The episode terminates automatically with a negative reward if the agent does not produce a labeling action before a maximum time limit is reached.

Crucially, the transition between interaction and labeling does not happen at a fixed time, but is initiated by the agent. This is achieved by providing the agent with the ability to produce either an interaction action or a labeling action at every time step. This allows the agent to decide when enough information has been gathered, and forces it to balance the trade-off between answering now given its current knowledge, or delaying its answer to gather more information.

The optimal trade-off between information gathering and risk of answering incorrectly depends on two factors. The first factor is the *difficulty* of the question and the second is the *cost of information*. The difficulty is environment specific and is addressed later when we describe the environments. The cost of information can be generically controlled by varying the discount factor during learning. A small discount factor places less emphasis on future rewards and encourages the agent to answer as quickly as possible. On the other hand, a large discount factor encourages the agent to spend more time gathering information in order to increase the likelihood of choosing the correct answer.

Our use of "questions" and "answers" differs from how these terms are used elsewhere in the literature. Sutton et al. (2011) talk about a value function as a question, and the agent provides an answer in the form of an *approximation* of the value. The answer incorporates the agent's *knowledge*, and the match between the actual value and the agent's approximation grounds what it means for this knowledge to be accurate.

In our usage the environment (or episode) itself is the question, and answers come in the form of labeling actions. In each episode there is a *correct* answer whose semantics is grounded in the sign of the reward function, and the accuracy of an agents knowledge is assessed by the frequency with which it is able to choose the correct answer.

Using reward (rather than value) to ground our semantics means that we have a straightforward way to ask questions that do not depend on the agent's behavior. For example, we can easily ask the question "Which block is heaviest?" without making the question contingent on a particular information acquisition strategy.

## 4  AGENT ARCHITECTURE AND TRAINING

We use the same basic agent architecture and training procedure for all of our experiments, making only minimal modifications in order to adapt the agents to different observation spaces and actuators. For all experiments we train recurrent agents using an LSTM with 100 hidden units. When working from features we feed the observations into the LSTM directly. When training from pixels we first scale the observations to 84x84 pixels and feed them through a three convolution layers, each

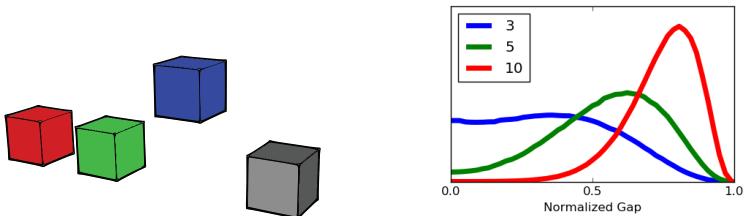

Figure 1: **Left:** Diagram of the Which is Heavier environment. Blocks are always arranged in a line, but mass of the different blocks changes from episode to episode. **Right:** Mass gap distributions for different settings of $\beta$ used in the experiments.

followed by a ReLU non-linearity. The three layers have 32, 64, 64 square filters with sizes 8, 4, 3, which are applied at strides of 4, 2, 1 respectively. We train the agents using Asynchronous Advantage Actor Critic (Mnih et al., 2016), but ensure that the unroll length is always greater than the timeout length so the agent network is unrolled over the entirety of each episode.

# 5 WHICH IS HEAVIER

The Which is Heavier environment is designed to ask a question about the relative masses of different objects in a scene. We assign masses to objects in a way that is uncorrelated with their appearance in order to ensure that the task is not solvable without interaction.

## 5.1 ENVIRONMENT

The environment is diagrammed in the left panel of Figure 1. It consists of four blocks, which are constrained to only move vertically. The blocks are always the same size, but vary in mass between episodes. The agent's strength (i.e. magnitude of force it can apply) remains constant between episodes.

The question to answer in this environment is which of the four blocks is the heaviest. Since the mass of each block is randomly assigned in each episode, the agent must poke the blocks and observe how they respond in order to make this determination. Assigning masses randomly ensures it is not possible to solve this task from vision (or features) alone, since the appearance and identity of each block imparts no information about its mass in the current episode. The only way to obtain information about the masses of the blocks is to interact with them and watch how they respond.

The Which is Heavier environment is designed to encode a latent bandit problem through a "physical" lens. Each block corresponds to an arm of the bandit, and the reward obtained by pulling each arm is proportional to the mass of the block. Identifying the heaviest block can then be seen as a best arm identification problem (Audibert & Bubeck, 2010). Best arm identification is a well studied problem in experimental design, and understanding of how an optimal solution to the latent bandit should behave is used to guide our analysis of the agents we train on this task.

It is important to emphasize that we cannot simply apply standard bandit algorithms here, because we impose a much higher level of prior ignorance on our algorithms than that setting allows. Bandit algorithms assume that rewards are observed directly, whereas our agents observe mass through its role in dynamics (and in the case of learning from pixels, through the lens of vision as well). To maintain a bandit setting one could imagine parameterizing this transformation from reward to observation, and perhaps even learning the mapping as well; however, doing so requires explicitly acknowledging the mapping in the design of the learning algorithm, which we avoid doing. More-over, acknowledging this mapping in any way requires the a-priori recognition of the existence of the latent bandit structure. From the perspective of our learning algorithm the mere existence of such a structure also lies beyond the veil of ignorance.

Controlling the distribution of masses allows us to control the difficulty of this task. In particular, by controlling the size of the mass gap between the two heaviest blocks we can make the task more

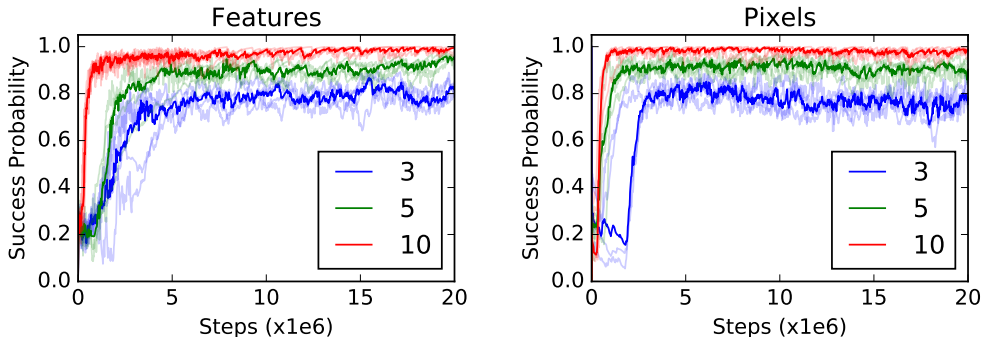

Figure 2: Learning curves for a typical agent trained on the Which is Heavier environment at varying difficulty settings. The y-axes show the probability of the agent producing the correct answer before the episode times out. Each plot shows the top 50% of agents started from 10 random seeds with identical hyperparameter settings. The light lines show learning curves from individual agents, and the dark lines show the median performance across the displayed runs for each difficulty. **Left:** Agents trained from features. **Right:** Agents trained from pixels.

or less difficult. We generate masses in the range $[0, 1]$ and scale them to an appropriate range for the agent's strength.

We use the following scheme for controlling the difficulty of the Which is Heavier environment. First we select one of the blocks uniformly at random to be the "heavy" block and designate the remaining three as "light" blocks. We sample the mass of the heavy block from $\mathrm{Beta}(\beta, 1)$ and the mass of the light blocks from $\mathrm{Beta}(1, \beta)$. The single parameter $\beta$ effectively controls the distribution of mass gaps (and thus controls the difficulty), with large values of $\beta$ leading to easier problems. Figure 1 shows the distribution of mass gaps for three values of $\beta$ that we use in our experiments.

We distinguish between *problem* level and *instance* level difficulty for this domain. Instance level difficulty refers to the size of the mass gap in a single episode. If the mass gap is small it is harder to determine which block is heaviest, and we say that one episode is more difficult than another by comparing their mass gaps. Problem level difficulty refers to the shape of the generating distribution of mass gaps (e.g. as shown in the right panel of Figure 1). A distribution that puts more mass on configurations that have a small mass gap will tend to generate more episodes that are difficult at the instance level, and we say that one distribution is more difficult than another if it is more likely to generate instances with small mass gaps. We control the problem level difficulty through $\beta$, but we incorporate both problem and instance level difficulty in our analysis.

We set the episode length limit to 100 steps in this environment, which is sufficient time to be much longer than a typical episode by a successfully trained agent.

## 5.2 ACTUATORS

The obvious choice for actuation in physical domains is some kind of arm or hand based manipulator. However, controlling an arm or hand is quite challenging on its own, requiring a fair amount of dexterity on the part of the agent. The manipulation problem, while very interesting in its own right, is orthogonal to our goals in this work. Therefore we avoid the problem of learning dexterous manipulation by providing the agent with a much simpler form of actuation.

We call the actuation strategy for this environment *direct* actuation, which allows the agent to affect forces on the different blocks directly. At every time step the agent can output one out of eight possible actions. The first four actions result in an application of a vertical force of fixed magnitude to center of mass of each of the four blocks respectively. The remaining actions are labeling actions and correspond to agent's selection of which is the heaviest block.

## 5.3 Experiments

Our first experiment is a sanity check to show that we can train agents successfully on the Which is Heavier environment using both features and pixels. This experiment is designed simply to show that our task is solvable, and to illustrate that by changing the problem difficulty we can make the task very hard.

We present two additional experiments showing how varying difficulty leads to differentiated behavior both at the problem level and at the instance level. In both cases knowledge of the latent bandit problem allows us to make predictions about how an experimenting agent should behave, and our experiments are designed to show that qualitatively correct behavior is obtained by our agents in spite of their a-priori ignorance of the underlying bandit problem.

We show that as we increase the problem difficulty the learned policies transition from guessing immediately when a heavy block is found to strongly preferring to poke all blocks before making a decision. This corresponds to the observation that if it is unlikely for more than one arm to give high reward then any high reward arm is likely to be best.

We also observe that our agents can adapt their behavior to the difficulty of individual problem instances. We show that a single agent will tend to spend longer gathering information when the particular problem instance is more difficult. This corresponds to the observation that when the two best arms have similar reward then more information is required to accurately distinguish them.

Finally, we conduct an experiment comparing our learned information gathering policies to a randomized baseline method. This experiment shows that agents more reliably produce the correct label by following their learned interaction policies than by observing the environment being driven by random actions.

**Success in learning**  For this experiment we trained several agents at three different difficulties corresponding to $\beta \in \{3, 5, 10\}$. For each problem difficulty we trained agents on both feature observations, which includes the $z$ coordinate of each of the four blocks; and also using raw pixels, providing $84 \times 84$ pixel RGB rendering of the scene to the agent. Representative learning curves for each condition are shown in Figure 2. The curves are smoothed over time and show a running estimate of the probability of success, rather than showing the reward directly.

The agents do not reach perfect performance on this task, with more difficult problems plateauing at progressively lower performance. This can be explained by looking at the distributions of instance level difficulties generated by different settings of $\beta$, which is shown in the right panel of Figure 1. For higher difficulties (lower values of $\beta$) there is a substantial probability of generating problem instances where the mass gap is near 0, which makes distinguishing between the two heaviest blocks very difficult.

**Population strategy differentiation**  For this experiment we trained agents at three different difficulties corresponding to $\beta \in \{3, 5, 10\}$ all using a discount factor of $\gamma = 0.95$ which corresponds a relatively high cost of gathering information. We trained three agents for each difficulty and show results aggregated across the different replicas.

After training, each agent was run for 10,000 steps under the same conditions they were exposed to during training. We record the number and length of episodes executed during the testing period as well as the outcome of each episode. Episodes are terminated by timeout after 100 steps, but the vast majority of episodes are terminated in $< 30$ steps by the agent producing a label. Since episodes vary in length not all agents complete the same number of episodes during testing.

The left plot in Figure 3 shows histograms of the episode lengths broken down by task difficulty. The dashed vertical line indicates an episode length of four interaction steps, which is the minimum number of actions required for the agents to interact with every block. At a task difficulty of $\beta = 10$ the agents appear to learn simply to search for a single heavy block (which can be found with an average of two interactions). However, at a task difficulty of $\beta = 3$ we see a strong bias away from terminating the episode before taking at least four exploratory actions.

**Individual strategy differentiation**  For this experiment we trained agents using the same three task difficulties as in the previous experiment, but with an increased discount factor of $\gamma = 0.99$.

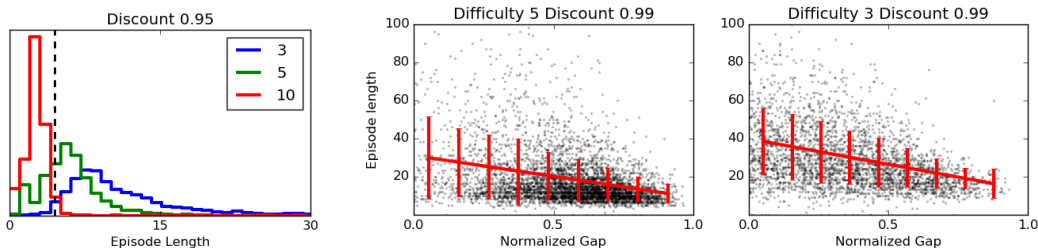

Figure 3: **Left:** Histograms of episode lengths for different task difficulty ($\beta$) settings. There is a transition from $\beta = 10$ where the agents answer eagerly as soon as they find a heavy block to $\beta = 3$ where the agents are more conservative about answering before they have acted enough to poke all the blocks at least once. **Right:** Episode lengths as a function of the normalized mass gap. Units on the x-axes are scaled to the range of possible masses, and the y-axis shows the number of steps before the agent takes a labeling action. The black dots show individual episodes, and the red line shows a linear trend fit by OLS and error bars show a histogram estimate of standard deviations. Each plot shows the testing episodes of a single trained agent.

This decreases the cost of exploration and encourages the agents to gather more information before producing a label, leading to longer episodes.

After training, each agent was run for 100,000 steps under the same conditions they were exposed to during training. We record the length of each episode, as well as the mass gap between the two heaviest blocks in each episode. In the same way that we use the distribution of mass gaps as a measure of task difficulty, we can use the mass gap in a single episode as a measure of the difficulty of that specific problem instance. We again exclude from analysis the very small proportion of episodes that terminate by timeout.

The right plots in Figure 3 show the relationship between the mass gap and episode length across the testing runs of two different agents. From these plots we can see how a single agent has learned to adapt its behavior based on the difficulty of a single problem instance. Although the variance is high, there is a clear correlation between the mass gap and the length of the episodes. This behavior reflects what we would expect from a solution to the latent bandit problem; more information is required to identify the best arm when the second best arm is nearly as good.

**Randomized interaction**   For this experiment we trained several agents using both feature and pixel observations at the same three task difficulties with a discount of $\gamma = 0.95$. In total we trained six sets of agents for this experiment.

After training, each agent was run for 10,000 steps under the same conditions used during training. We record the outcome of each episode, as well as the number of steps taken by each agent before it chooses a label. For each agent we repeat the experiment using both the agent's learned interaction policy as well as a *randomized interaction* policy.

The randomized interaction policy is obtained as follows: At each step the agent chooses a *candidate action* using its learned policy. If the candidate action is a labeling action then it is passed to the environment unchanged (and the episode terminates). However, if the candidate action is an interaction action then we replace the agent action with a new interaction action chosen uniformly at random from the available action set. When following the randomized interaction policy the agent has no control over the information gathering process, but still controls when each episode ends, and what label is chosen.

Figure 4 compares the learned interaction policies to the randomized interaction baselines. The results show that the effect on episode length is small, with no consistent bias towards longer or shorter episodes across difficulties and observation types. However, the learned interaction policies produce more accurate labels across all permutations.

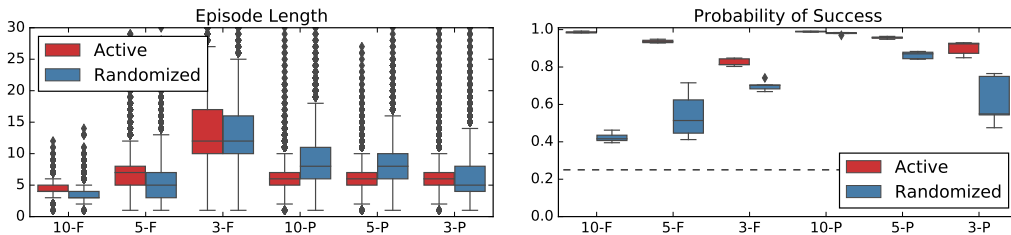

Figure 4: Comparison between agents in the Which is Heavier environment following their learned interaction policies vs the randomized interaction policy baseline. The x-axes show Difficulty-Observation combinations (e.g. 10-F is difficulty 10 with feature observations and 3-P is difficulty 3 with pixel observations) **Left:** Episode lengths when gathering information using the different interaction policies. **Right:** Probability of choosing the correct label under different conditions (episodes terminating in timeout have been excluded). The dashed line shows chance performance.

# 6 TOWERS

The Towers environment is designed to ask agents to count the number of cohesive rigid bodies in a scene. The environment is designed so that in its initial configuration it is not possible to determine the number of rigid bodies from vision or features alone.

## 6.1 ENVIRONMENT

The environment is diagrammed in the left panel of Figure 5. It consists of a tower of five blocks which can move freely in three dimensions. The initial block tower is always in the same configuration but in each episode we bolt together different subsets of the blocks to form larger rigid bodies as shown in the figure.

The question to answer in this environment is how many rigid bodies are formed from the primitive blocks. Since which blocks are bound together is randomly assigned in each episode, and binding forces are invisible, the agent must poke the tower and observe how it falls down in order to determine how many rigid bodies it is composed of. We parameterize the environment in such a way that the distribution over the number of separate blocks in the tower is uniform. This ensures that there is no single action strategy that achieves high reward.

## 6.2 ACTUATORS

In the Towers environment, we used two actuators: *direct* actuation, which is similar to the Which is Heavier environment; and the *fist* actuator, described below. In case of the direct actuation, the agent can output one out of 25 actions. At every time step, the agent can apply a force of fixed magnitude in either of +x, -x, +y or -y direction to one out of the five blocks. If two blocks are glued together, both blocks move under the effect of force. We use towers of five blocks, which results in 20 different possible actions. The remaining actions are labeling actions that are used by the agent to indicate the number of distinct blocks in the tower.

The fist is a large spherical object that the agent can actuate by setting velocities in a 2D horizontal plane. Unlike direct actuation, the agent cannot apply any direct forces to the objects that constitute the tower, but only manipulate them by pushing or hitting them with the fist. At every time step agent can output one of nine actions. The first four actions corresponds to setting the velocity of the fist to a constant amount in (+x, -x, +y, -y) directions respectively. The remaining actions are labeling actions, that are used by the agent to indicate the number of distinct blocks in the tower.

In order to investigate if the agent learns a strategy of stopping after a fixed number of time steps or whether it integrates sensory information in a non-trivial manner we used a notion of "control time step". The idea of control time step is similar to that of action repeats and if the physics simulation time step is 0.025s and control time step is 0.1s, it means that the same action is repeated 4 times. For the direct actuators we use an episode timeout of 26 steps and for both actuator types.

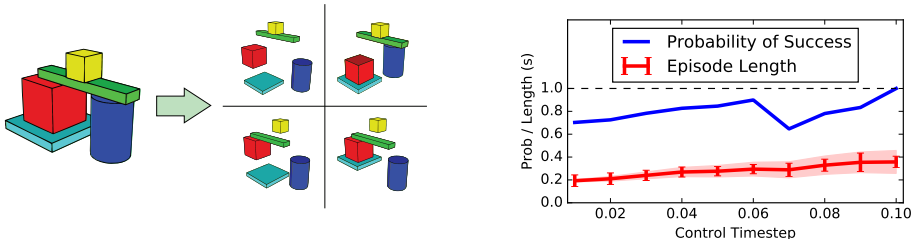

Figure 5: **Top:** Example trajectory of a block tower being knocked down using the fist actuator. **Left:** Diagram of the hidden structure of the Towers environment. The tower on the left is composed of five blocks, but could decompose into rigid objects in any several ways that can only be distinguished by interacting with the tower. **Right:** Behavior of a single trained agent using fist actuators when varying the control time step. The x-axis shows different control time step lengths (the training condition 0.1). The blue line shows probability of the agent correctly identifying the number of blocks. The red line shows the median episode length (in seconds) with error bars showing 95% confidence intervals computed over 50 episodes. The shaded region shows $+/-1$ control time step around the median.

## 6.3 EXPERIMENTS

Our first experiment is again intended to show that we can train agents in this environment. We show simply that the task is solvable by our agents using both types of actuation.

The second experiment shows that the agents learn to wait for an observation where they can identify the number of rigid bodies before producing an answer. This is designed to show that the agents find a closed loop strategy for counting the number of rigid bodies. An alternative hypothesis would be that agents learn to wait for (approximately) the same number of steps each time and then take their best guess.

Our third experiment compares the learned policy to a randomized interaction policy and shows that agents are able to determine the correct number of blocks in the tower more quickly and more reliably when using their learned policy to gather information.

**Success in learning**    For this experiment we trained several agents on the Towers environment using different pairings of actuators and perception. The features observations include the 3d position of each primitive block, and when training using raw pixels we provide an $84 \times 84$ pixel RGB rendering of the scene as the agent observation. Figure 6 shows learning curves for each combination of actuator and observation type.

In all cases we obtain agents that solve the task nearly perfectly, although when training from pixels we find that the range of hyperparameters which train successfully is narrower than when training from features. Interestingly, the fist actuators lead to the fastest learning, in spite of the fact that the agent must manipulate the blocks indirectly through the fist. One possible explanation is that the fist can affect multiple blocks in one action step, whereas in the direct actuation only one block can be affected per time step.

**Waiting for information**    For this experiment we trained an agent with pixel observations and the fist actuator on the towers task with an control time step of 0.1 seconds and examine its behavior at test time with a smaller delay between actions. Reducing the control time step means that from the agent perspective time has been slowed down. Moving the fist a fixed amount of distance takes longer, as does waiting for the block tower to collapse once it has been hit.

After training the agent was run for 10000 steps for a range of different control time steps. We record the outcome of each episode, as well as the number of steps taken by the agent before it chooses a label. None of the test episodes terminate by timeout, so we include all of them in the analysis.

The plot in Figure 5 shows the probability of answering correctly, as well as the median length of each episode measured in seconds. In terms of absolute performance we see a small drop compared to the training setting, where the agent is essentially perfect, but the agent performance remains good even for substantially smaller control timesteps than were used during training.

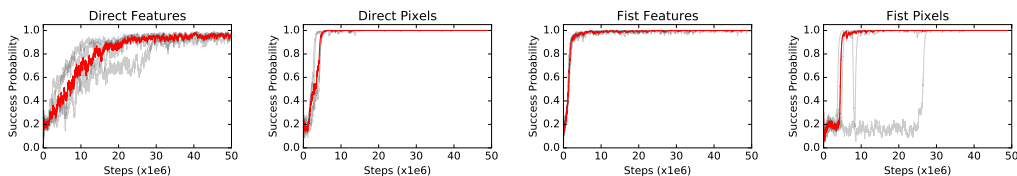

Figure 6: Learning curves for agents trained on the Towers environment under different conditions. The y-axes show the probability of the agent producing the correct answer before the episode times out. The different plots show different pairings of observations and actuators as indicated in the plot titles. Each plot shows the top 50% of runs from 10 random seeds with identical hyper-parameter settings. The black lines show learning curves from individual agents, and the red lines show the median performance of the displayed runs.

We also observe that the episodes with different time steps take approximate the same amount of real time across the majority of the tested range. This corresponds to a large change in episode length as measured by number of agent actions, since with an control time step of 0.01 the agent must execute 10x as many actions to cover the same amount of real time as compared to the control time step used during training. From this we can infer that the agent has learned to wait for an informative observation before producing a label, as opposed to a simpler degenerate strategy of waiting a fixed amount of steps before answering.

**Randomized interaction** For this experiment we trained several agents for each combination of actuator and observation type, and examine their behavior when observing an environment driven by a random interaction policy. The randomized interaction policy is identical to the randomized baseline used in the Which is Heavier environment.

After training, each agent was run for 10,000 steps. We record the outcome of each episode, as well as the number of steps taken by the agent before it chooses a label. For each agent we repeat the experiment using both the agent's learned interaction policy as well as the randomized interaction policy.

Figure 7 compares the learned interaction policies to the randomized interaction baselines. The results show that the agents tend to produce labels more quickly when following their learned interaction policies, and also that the labels they produce in this way are much more accurate.

# 7 RELATED WORK

Deep learning techniques in conjunction with vast labeled datasets have yielded powerful models for image classification (Krizhevsky et al., 2012; He et al., 2016) and speech recognition (Hinton et al., 2012). In recent years, as we have approached human level performance on these tasks, there has been a strong interest in the computer vision field in moving beyond semantic classification, to tasks that require a deeper and more nuanced understanding of the world.

Inspired by developmental studies (Smith & Gasser, 2005), some recent works have focused on learning representations by predicting physical embodiment quantities such as ego-motion (Agrawal et al., 2015; Jayaraman & Grauman, 2015), instead of symbolic labels. Extending the realm of things-to-be-predicted to include quantities beyond class labels, such as viewer centric parameters (Doersch et al., 2015) or the poses of humans within a scene (Delaitre et al., 2012; Fouhey et al., 2014), has been shown to improve the quality of feature learning and scene understanding. Researchers have looked at cross modal learning, for example synthesizing sounds from visual images (Owens et al., 2015), using summary statistics of audio to learn features for object recognition (Owens et al., 2016) or image colorization (Zhang et al., 2016).

Inverting the prediction tower, another line of work has focused on learning about the visual world by synthesizing, rather than analyzing, images. Major cornerstones of recent work in this area include the Variational Autoencoders of Kingma & Welling (2014), the Generative Adversarial Networks

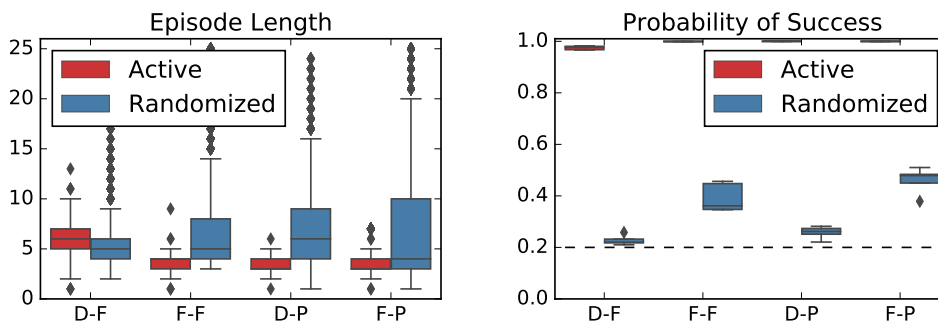

Figure 7: Comparison between agents in the Towers environment following their learned interaction policies vs the randomized interaction policy baseline. The x-axes show different Observation-Actuator combinations (e.g. D-F is Direct-Features and F-P is Fist-Pixels). **Left:** Episode lengths when gathering information using the different interaction policies. **Right:** Probability of choosing the correct label under different conditions (episodes terminating in timeout have been excluded). The dashed line shows chance performance.

of (Goodfellow et al., 2014), and more recently autoregressive models have been very successful (van den Oord et al., 2016).

Building on models of single image synthesis there have been many works on predicting the evolution of video frames over time (Ranzato et al., 2014; Srivastava et al., 2015; van den Oord et al., 2016). Xue et al. (2016) have approached this problem by designing a variational autoencoder architecture that uses the latent stochastic units of the VAE to make choices about the direction of motion of objects, and generates future frames conditioned on these choices.

A different form of uncertainty in video prediction can arise from the effect of actions taken by an agent. In environments with deterministic dynamics (where the possibility of "known unknowns" can, in principle, be eliminated), very accurate action-conditional predictions of future frames can be made (Oh et al., 2015). Introducing actions into the prediction process amounts to learning a latent forward dynamics model, which can be exploited to plan actions to achieve novel goals (Watter et al., 2015; Assael et al., 2015; Fragkiadaki et al., 2016). In these works, frame synthesis plays the role of a regularizer, preventing collapse of the feature space where the dynamics model lives.

Agrawal et al. (2016) break the dependency between frame synthesis and dynamics learning by replacing frame synthesis with an *inverse* dynamics model. The forward model plays the same role as in the earlier works, but here feature space collapse is prevented by ensuring that the model can decode actions from pairs of time-adjacent images. Several works, including Agrawal et al. (2016) and Assael et al. (2015) mentioned above but also Pinto et al. (2016); Pinto & Gupta (2016); Levine et al. (2016), have gone further in coupling feature learning and dynamics. The learned dynamics models can be used for control not only after learning but also during the learning process in order to collect data in a more targeted way, which has been shown to improve the speed and quality of learning in robot manipulation tasks.

A key challenge of learning from dynamics is collecting the appropriate data. An ingenious solution to this is to import real world data into a physics engine and simulate the application of forces in order to generate ground truth data. This is the approach taken by Mottaghi et al. (2016), who generate an "interactable" data set of scenes, which they use to generate a static data set of image and force pairs, along with the ground truth trajectory of a target object in response to the application of the indicated force.

When the purpose is learning an intuitive understanding of dynamics it is possible to do interesting work with entirely synthetic data (Fragkiadaki et al., 2016; Lerer et al., 2016). Lerer et al. (2016) show that convolutional networks can learn to make judgments about the stability of synthetic block towers based on a single image of the tower. They also show that their model trained on synthetic data is able to generalize to make accurate judgments about photographs of similar block towers built in the real world.

Making intuitive judgments about block towers has been extensively studied in the psychophysics literature. There is substantial evidence connecting the behavior of human judgments to inference over an explicit latent physics model (Hegarty, 2004; Hamrick et al., 2011; Battaglia et al., 2013). Humans can infer mass by watching movies of complex rigid body dynamics (Hamrick et al., 2016).

A major component of the above line of work is *analysis by synthesis*, in which understanding of a physical process is obtained by learning to invert it. Observations are assumed to be generated from an explicitly parameterized generative model of the true physical process, and provide constraints to an inference process run over the parameters of this model. The analysis by synthesis approach has been extremely influential due to its power to explain human judgments and generalization patterns in a variety of situations (Lake et al., 2015).

Galileo (Wu et al., 2015) is a particularly relevant instance of tying together analysis by synthesis and deep learning for understanding dynamics. This system first infers the physical parameters (mass and friction coefficient) of a variety of blocks by watching videos of them sliding down slopes and colliding with other blocks. This stage of the system uses an off-the-shelf object tracker to ground inference over the parameters of a physical simulator, and the inference is achieved by matching simulated and observed block trajectories. The inferred physical parameters are used to train a deep network to predict the physical parameters from the initial frame of video. At test time the system is evaluated by using the deep network to infer physical parameters of new blocks, which can be fed into the physics engine and used to answer questions about behaviors not observed at training time.

Physics 101 (Wu et al., 2016) is an extension of Galileo that more fully embraces deep learning. Instead of using a first pass of analysis by synthesis to infer physical parameters based on observations, a deep network is trained to regress the output of an object tracker directly, and the relevant physical laws are encoded directly into the architecture of the model. The authors show that they can use latent intrinsic physical properties inferred in this way to make novel predictions. The approach of encoding physical models as architecture constraints has also been proposed by Stewart & Ermon (2016).

Many of the works discussed thus far, including Galileo and Physics 101, are restricted to passive sensing. Pinto et al. (2016); Pinto & Gupta (2016); Agrawal et al. (2016); Levine et al. (2016) are exceptions to this because they learn their models using a sequential greedy data collection bootstrapping strategy. Active sensing, it appears, is an important aspect of visual object learning in toddlers as argued by Bambach et al. (2016), providing motivation for the approach presented here.

In computer vision, it is well known that recognition performance can be improved by moving so as to acquire new views of an object or scene. Jayaraman & Grauman (2016), for example, apply deep reinforcement learning to construct an agent that chooses how to acquire new views of an object so as to classify it into a semantic category, and their related work section surveys many other efforts in active vision.

While Jayaraman & Grauman (2016) and others share deep reinforcement learning and active sensing in common with our work, their goal is to learn a policy that can be applied to images to make decisions based on vision. In contrast, the goal in this paper is to study how agents learn to experiment continually so as to learn representations to answer questions about intrinsic properties of objects. In particular, our focus is on tasks that can only be solved by interaction and not by vision alone.

## 8 CONCLUSION AND FUTURE DIRECTIONS

Despite recent advances in artificial intelligence, machines still lack a common sense understanding of our physical world. There has been impressive progress in recognizing objects, segmenting object boundaries and even describing visual scenes with natural language. However, these tasks are not enough for machines to infer physical properties of objects such as mass, friction or deformability.

We introduce a deep reinforcement learning agent that actively interacts with physical objects to infer their hidden properties. Our approach is inspired by findings from the developmental psychology literature indicating that infants spend a lot of their early time experimenting with objects through random exploration (Smith & Gasser, 2005; Gopnik, 2012; Spelke & Kinzler, 2007). By letting our agents conduct physical experiments in an interactive simulated environment, they learn to manip-

ulate objects and observe the consequences to infer hidden object properties. We demonstrate the efficacy of our approach on two important physical understanding tasks—inferring mass and counting the number of objects under strong visual ambiguities. Our empirical findings suggest that our agents learn different strategies for these tasks that balance the cost of gathering information against the cost of making mistakes in different situations.

Scientists and children are able not only to probe the environment to discover things about it, but they can also leverage their findings to answer new questions. In this paper we have shown that agents can be trained to gather knowledge to answer questions about hidden properties, but we have not addressed the larger issue of theory building, or transfer of this information. Given agents that can make judgments about mass and numerosity, how can they be enticed to leverage this knowledge to solve new tasks?

Another important aspect of understanding through interaction is that that the shape of the interactions influences behavior. We touched on this in the Towers environment where we looked at two different actuation styles, but there is much more to be done here. Thinking along these lines leads naturally to exploring tool use. We showed that agents can make judgments about object mass by hitting them, but could we train an agent to make similar judgments using a scale?

Finally, we have made no attempt in this work to optimize data efficiency, but learning physical properties from fewer samples is an important direction to pursue.

ACKNOWLEDGMENTS

We would like to thank Matt Hoffman for several enlightening discussions about bandits. We would also like to thank the ICLR reviewers, whose helpful feedback allowed us to greatly improve the paper.

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
