# Peer review of "Learning to Perform Physics Experiments via Deep Reinforcement Learning"

_ICLR 2017 — accepted_

[Official Review · AnonReviewer5 · rating 7 · confidence 3 · 16 Dec 2016]
**Conceptually interesting and valuable, but lacks commitment to precise definitions**

This paper investigates the question of gathering information (answering question)
through direct interaction with the environment. In that sense, it is closely
related to "active learning" in supervised learning, or to the fundamental
problem of exploration-exploitation in RL. The authors consider a specific 
instance of this problem in a physics domain and learn
information-seeking policies using recent deep RL methods.

The paper is mostly empirical and explores the effect of changing the
cost of information (via the discount factor) on the structure of the learned
policies. It also shows that general-purpose deep policy gradient methods are
sufficient powerful to learn such tasks. The proposed environment is, to my knowledge,
novel as well the task formulation in section 2. (And it would be very valuable to the
the community if the environment would be open-sourced)

The expression "latent structure/dynamics" is used throughout the text and the connection
with bandits is mentioned in section 4. It therefore seems that authors aspire
for more generality with their approach but the paper doesn't quite fully ground
the proposed approach formally in any existing framework nor does it provide a
new one completely.

For example: how does your approach formalize the concept of "questions" and "answers" ?
What makes a question "difficult" ? How do you quantify "difficulty" ?
How do you define the "cost of information"? What are its units (bits, scalar reward), its semantics ?
Do you you have an MDP or a POMDP ? What kind of MDP do you consider ?
How do you define your discounted MDP ? What is the state and action spaces ?
Some important problem structure under the "interaction/labeling/reward"
paragraph of section 2 would be worth expressing directly in your definition
of the MDP: labeling actions can only occur during the "labeling phase" and that the transition
and reward functions have a specific structure (positive/negative, lead to absorbing state).
The notion of "phase" could perhaps be implemented by considering an augmented state space : $\tilde s = (s, phase)$

[Official Review · AnonReviewer6 · rating 7 · confidence 3 · 16 Dec 2016 (modified: 18 Jan 2017)]

This paper purports to investigate the ability of RL agents to perform ‘physics experiments’ in an environment, to infer physical properties about the objects in that environment. The problem is very well motivated; indeed, inferring the physical properties of objects is a crucial skill for intelligent agents, and there has been relatively little work in this direction, particularly in deep RL. The paper is also well-written.

As there are no architectural or theoretical contributions of the paper (and none are claimed), the main novelty comes in the task application – using a recurrent A3C model for two tasks that simulate an agent interacting with an environment to infer physical properties of objects. More specifically, two tasks are considered – moving blocks to determine their mass, and poking towers such that they fall to determine the number of rigid bodies they are composed of. These of course represent a very limited cross-section of the prerequisite abilities for an agent to understand physics. This in itself is not a bad thing, but since there is no comparison of different (simpler) RL agents on the tasks, it is difficult to determine if the tasks selected are challenging. As mentioned in the pre-review question, the ‘Which is Heavier’ task seems quite easy due to the actuator set-up, and the fact that the model simply must learn to take the difference between successive block positions (which are directly encoded as features in most experiments).  Thus, it is not particularly surprising that the RL agent can solve the proposed tasks. 

The main claim beyond solving two proposed tasks related to physics simulation is that “the agents learn different strategies for these tasks that balance the cost of gathering information against the cost of making mistakes”. The ‘cost of gathering information’ is implemented by multiplying the reward with a value of gamma < 1. This is somewhat interesting behaviour, but is hardly surprising given the problem setup.

One item the authors highlight is that their approach of learning about physical object properties through interaction is different from many previous approaches, which use visual cues. However, the authors also note that this in itself is not novel, and has been explored in other work (e.g. Agrawal et al. (2016)). I think it’s crucial for the authors to discuss these approaches in more detail (potentially along with removing some other, less relevant information from the related work section), and specifically highlight why the proposed tasks in this paper are interesting compared to, for example, learning to move objects towards certain end positions by poking them.

To discern the level of contribution of the paper, one must ask the following questions: 

1)	how much do these two tasks contribute (above previous work) to the goal of having agents learn the properties of objects by interaction; and
2)	how much do the results of the RL agent on these tasks contribute to our understanding of agents that interact with their environment to learn physical properties of objects? 

It is difficult to know exactly, but due to the concerns outlined above, I am not convinced that the answers to (1) or (2) are “to a significant extent”. In particular, for (1), since the proposed agent is able to essentially solve both tasks, it is not clear that the tasks can be used to benchmark more advanced agents (e.g. it can’t be used as a set of bAbI-like tasks). 

Another possible concern, as pointed out by Reviewer 3, is that the description of the model is extremely concise. It would be nice to have, for example, a diagram illustrating the inputs and outputs to the model at each time step, to ease replication.

Overall, it is important to make progress towards agents that can learn to discover physical properties of their environment, and the paper contributes in this direction. However, the technical contributions of this paper are rather limited – thus, it is not clear to what extent the paper pushes forward research in this direction beyond previous work that is mentioned. It would be nice, for example, to have some discussion about the future of agents that learn physics from interaction (speculation on more difficult versions of the tasks in this paper), and how the proposed approach fits into that picture.  

---------------
EDIT: score updated, see comments below

[Official Review · AnonReviewer3 · rating 7 · confidence 4 · 16 Dec 2016 (modified: 18 Jan 2017)]
**mixed opinion**
impact 4

This paper presents interesting experimental findings that state-of-the-art deep reinforcement learning methods enable agent learning of latent (physical) properties in its environment. The paper formulates the problem of an agent labeling environmental properties after interacting with the environment based on its actions, and applies the deep reinforcement learning model to evaluate whether such learning is possible. The approach jointly learns the convolutional layers for pixel-based perception and its later layers for learning actions based on reinforcement signals.

We have a mixed opinion about this paper. The paper is written clearly and presents interesting experimental findings. It introduces and formulates a problem potentially important for many robotics applications. Simultaneously, the paper suffers from lacking algorithmic contributions and missing (some of) crucial experiments to confirm its true benefits.

Pros:

+ This paper introduces a new problem of learning latent properties in the agent's environment.

+ The paper presents a framework to appropriately combine existing tools to address the formulated problem.

+ The paper tries reinforcement learning with image inputs and fist-like actuator actions. This will lead to its direct application to robots.

Cons:

- Lacking algorithmic contribution: this paper applies existing tools/methods to solve the problem rather than developing something new or extending them. The approach essentially is training LSTMs with convolutional layers using the previous Asynchronous Advantage Actor Critic.

- In the Towers experiment, the results of probably the most important setting, "Fist Pixels", are missing. This setting receiving pixel inputs and using the Fist actuator in a continuous space is the setting closest to real-world robots, and thus is very important to confirm whether the proposed approach will be directly applicable to real-world robots. However, Figure 5 is missing the results with this setting. Is there any reason behind this?

- The paper lacks its comparison to any baseline methods. Without explicit baselines, it is difficult to see what the agent is really learning and what aspect of the proposed approach is benefitting the task. For instance, in the Towers task, how would an agent randomly pushing/hitting the tower (using 'Fist') a number of times and then passively observing its consequence to produce a label perform compared to this approach? That is, how would an approach with a fixed action policy (but with everything else) perform compared to the full deep reinforcement learning version?

[Official Review · AnonReviewer7 · rating 6 · confidence 3 · 31 Dec 2016]
**Interesting ideas while lack of the details**

This paper addresses the question of how to utilize physical interactions to answer questions about physical outcomes. This question falls into a popular stream in ML community -- understanding physics. The paper moved a step further and worked on experimental setups where there is no prior about the physical properties/rules and it uses a deep reinforcement learning (DRL) technique to address the problem. My overall opinion about this paper is: an interesting attempt and idea, yet without a clear contribution.

The experimental setups are quite interesting. The goal is to figure out which blocks are heavier or which blocks are glued together -- only by pushing and pulling objects around without any prior. The paper also shows reasonable performances on each task with detailed scenarios.

While these experiments and results are interesting, the contribution is unclear. My main question is: does this result bring us any new insight? While the scenarios are interesting and focused on physical experiments, this is not any more different (potentially easier) than learning from playing games (e.g. Atari). In other words, are the tasks really different from other typical popular DRL tasks? To this end, I would have been more excited if authors showed some more new insights or experiments on learned representations and etc. Currently, the paper only discusses the factual outcome. For example, it describes the experimental setup and how much performances an agent could achieve. The authors could probably dissect the learned representations further, or discuss how the experimental results are linked to the human behavior or physical properties/laws.

I am very in-between for my overall rating. I think the paper could have a deeper analysis. I however recommend the acceptance because of its merit of the idea.



The followings are some detailed questions (not directly impacting my overall rating):
(1) Page 2 "we assume that the agent has no prior knowledge about the physical properties of objects, or the laws of physics, and hence must interact with the objects in order to learn to answer questions about these properties.": why does one "must" interact with objects in order to learn about the properties? Can't we also learn through observation?

(2) Figure 1right is missing a Y-axis label.

(3) Page 3: A relating to bandit is interesting, but the formal approach is all based on DRL.

(4) Page 5 "which makes distinguishing between the two heaviest blocks very difficult": I am a bit confused why having a small mass gap makes the task harder (unless it's really close to 0). Shouldn't a machine be possible to distinguish even a pixel difference of speed? If not, isn't this just because of the network architecture?

(5) Page 5 "Since the agents exhibit similar performance using pixels and features we conduct the remaining experiments in this section using feature observations, since these agents are substantially faster to train.": How about at least showing a correlation of performances at the instance level (rather than average performances)? Even so, I think this is a bit of big conclusion.

(6) Throughout the papers, I felt that many conclusions (e.g. difficulty and etc) are based on a particularly chosen training distribution. For example, how does an agent really know when the instance is any more difficult? Doesn't this really depend on the empirically learned distribution of training samples (i.e. P(m_3 | m_1, m_2), where m_i indicates masses of object 1, 2, and 3)? In other words, does what's hard/easy matter much unless this is more thoroughly tested over various types of distributions?

(7) Any baseline approach?

[Author Response · Misha Denil · 16 Jan 2017]
**Why This Paper is Important**

The machine learning field has profited from hundreds of yearly publications that propose a new algorithm, which outperforms baseline algorithms and has a theoretical proof for some idealized form of the algorithm. This paper, however, is different. It does not propose a new algorithm. It incidentally uses an RL algorithm, but many other RL algorithms could have been used just as well. This paper is not about new models or algorithms.

So what is this paper about? It is an initial research step toward understanding objects and intuitive reasoning in physical worlds. Why is this important? Our best AI agents currently fail on simple control tasks and simple games, such as Montezuma’s Revenge, because when they look at a screen that has a ladder, a key and a skull they don’t immediately know that keys open doors, that skulls are probably hazardous and best avoided, that ladders allow us to defy gravity, etc. The understanding of physics, relations and objects enables children to solve seemingly simple problems that our best existing AI agents do not come close to begin to solve.   

Endowing our agents with knowledge of objects would help enormously with planning, reasoning and exploration. Yet, this is far from trivial. First, what is an object? It turns out this is not an easy question to answer and has baffled psychologists and philosophers. To simplify matters, in this paper we restrict our attention to rigid bodies. Staring at a thing in one’s hands is not enough to understand what it is. Vision or open loop perception in general is not enough. Trying to pull the thing apart to see whether it is a single entity might help. (Children do indeed love to tear things like paper apart to understand them.) Touching the thing to see what happens also helps --- perhaps it lights up and starts beeping. Further interaction might reveal that it enables us to remotely talk to someone else, to tweet, etc. Much of the knowledge gained is the result of interaction, that is perception by action. 

This paper is also about designing tasks to understand how agents acquire intuitive reasoning and experimentation strategies in physical worlds. An abundant body of evidence in psychology --- see eg the works of Gerd Gigerenzer

[Author Response · Misha Denil · 16 Jan 2017]
**Updated Environments**

We would like to highlight to all reviewers that the experiments have been updated substantially since the previous version of the paper.  In particular, we now include the missing results for Fist Pixels as requested by Reviewer 3, as well as additional experiments comparing our models to random baseline policies in both environments.

In the process of making these updates we have have made several minor changes to the environments themselves. These changes have made the tower environment harder to solve, but do not alter any conclusions drawn from the experiments.

[Final Decision · Program Chairs · 06 Feb 2017]
**ICLR committee final decision**

The following statement best summarizes the contribution: "This paper shows that model free RL methods can learn how to gather information about physical properties of objects, even when this information is not available to a passive observer, and use this information to make decisions." So this is not a paper about new theory or algorithms, but rather about solving the problem of acquiring knowledge about the physics of the world around us, which is important for many problems and helps explain human performance in many tasks. There are still some concerns about the depth-of-analysis of the paper, but on balance, it is seen as an unconventional but interesting paper. As per AnonReviewer6, the final version could still aim to better address "What should other researchers focus on if they are trying to build agents that can understand physics intuitively (building off this work)?"
 -- Area chair